# Improved Schemes for Episodic Memory-based Lifelong Learning

**Yunhui Guo** [†] [*]    **Mingrui Liu** [‡] [*]    **Tianbao Yang**[‡]    **Tajana Rosing** [†]

University of California, San Diego, CA [†]    University of Iowa, Iowa City, IA [‡]

yug185@eng.ucsd.edu, mingrui-liu@uiowa.edu, tianbao-yang@uiowa.edu, tajana@ucsd.edu

## Abstract

Current deep neural networks can achieve remarkable performance on a single task. However, when the deep neural network is continually trained on a sequence of tasks, it seems to gradually forget the previous learned knowledge. This phenomenon is referred to as *catastrophic forgetting* and motivates the field called lifelong learning. Recently, episodic memory based approaches such as GEM [1] and A-GEM [2] have shown remarkable performance. In this paper, we provide the first unified view of episodic memory based approaches from an optimization's perspective. This view leads to two improved schemes for episodic memory based lifelong learning, called MEGA-I and MEGA-II. MEGA-I and MEGA-II modulate the balance between old tasks and the new task by integrating the current gradient with the gradient computed on the episodic memory. Notably, we show that GEM and A-GEM are degenerate cases of MEGA-I and MEGA-II which consistently put the same emphasis on the current task, regardless of how the loss changes over time. Our proposed schemes address this issue by using novel loss-balancing updating rules, which drastically improve the performance over GEM and A-GEM. Extensive experimental results show that the proposed schemes significantly advance the state-of-the-art on four commonly used lifelong learning benchmarks, reducing the error by up to 18%. Implementation is available at: https://github.com/yunhuiguo/MEGA

## 1   Introduction

A significant step towards artificial general intelligence (AGI) is to enable the learning agent to acquire the ability of remembering past experiences while being trained on a continuum of tasks [3, 4, 5]. Current deep neural networks are capable of achieving remarkable performance on a single task [6]. However, when the network is retrained on a new task, its performance drops drastically on previously trained tasks, a phenomenon which is referred to as *catastrophic forgetting* [7, 8, 9, 10, 11, 12, 13, 14]. In stark contrast, the human cognitive system is capable of acquiring new knowledge without damaging previously learned experiences.

The problem of catastrophic forgetting motivates the field called lifelong learning [4, 11, 14, 15, 16, 17, 18, 19]. A central dilemma in lifelong learning is how to achieve a balance between the performance on old tasks and the new task [4, 7, 18, 20]. During the process of learning the new task, the originally learned knowledge will typically be disrupted, which leads to catastrophic forgetting. On the other hand, a learning algorithm biasing towards old tasks will interfere with the learning of the new task. Several lines of methods are proposed recently to address this issue. Examples include regularization based methods [4, 21, 22], knowledge transfer based methods [23] and episodic memory based methods [1, 2, 24]. Especially, episodic memory based methods such as *Gradient Episodic Memory* (GEM) [1] and *Averaged Gradient Episodic Memory* (A-GEM) [2] have shown remarkable performance. In episodic memory based methods, a small episodic memory is used for storing examples from old tasks to guide the optimization of the current task.

---

[*] Equal contribution.

In this paper, we present the first unified view of episodic memory based lifelong learning methods, including GEM [1] and A-GEM [2], from an optimization's perspective. Specifically, we cast the problem of avoiding catastrophic forgetting as an optimization problem with composite objective. We approximately solve the optimization problem using one-step stochastic gradient descent with the standard gradient replaced by the proposed *Mixed Stochastic Gradient* (MEGA). We propose two different schemes, called MEGA-I and MEGA-II, which can be used in different scenarios. We show that both GEM [1] and A-GEM [2] are degenerate cases of MEGA-I and MEGA-II which consistently put the same emphasis on the current task, regardless of how the loss changes over time. In contrast, based on our derivation, the direction of the proposed MEGA-I and MEGA-II balance old tasks and the new task in an adaptive manner by considering the performance of the model in the learning process.

Our contributions are as follows. (1) We present the first unified view of current episodic memory based lifelong learning methods including GEM [1] and A-GEM [2]. (2) From the presented unified view, we propose two different schemes, called MEGA-I and MEGA-II, for lifelong learning problems. (3) We extensively evaluate the proposed schemes on several lifelong learning benchmarks, and the results show that the proposed MEGA-I and MEGA-II significantly advance the state-of-the-art performance. We show that the proposed MEGA-I and MEGA-II achieve comparable performance in the existing setting for lifelong learning [2]. In particular, MEGA-II achieves an average accuracy of 91.21±0.10% on **Permuted MNIST**, which is 2% better than the previous state-of-the-art model. On **Split CIFAR**, our proposed MEGA-II achieves an average accuracy of 66.12±1.93%, which is about 5% better than the state-of-the-art method. (4) Finally, we show that the proposed MEGA-II outperforms MEGA-I when the number of examples per task is limited. We also analyze the reason for the effectiveness of MEGA-II over MEGA-I in this case.

## 2 Related Work

Several lifelong learning methods [25, 26] and evaluation protocols [27, 28] are proposed recently. We categorize the methods into different types based on the methodology,

**Regularization based approaches:** EWC [4] adopted Fisher information matrix to prevent important weights for old tasks from changing drastically. In PI [21], the authors introduced *intelligent synapses* and endowed each individual synapse with a local measure of "importance" to avoid old memories from being overwritten. RWALK [22] utilized a KL-divergence based regularization for preserving knowledge of old tasks. While in MAS [29] the importance measure for each parameter of the network was computed based on how sensitive the predicted output function is to a change in this parameter. [30] extended MAS for task-free continual learning. In [31], an approximation of the Hessian was employed to approximate the posterior after every task. Uncertainties measures were also used to avoid catastrophic forgetting [32]. [33] proposed methods based on *approximate Bayesian* which recursively approximate the posterior of the given data.

**Knowledge transfer based methods:** In PROG-NN [23], a new "column" with lateral connections to previous hidden layers was added for each new task. In [34], the authors proposed a method to leverage unlabeled data in the wild to avoid catastrophic forgetting using knowledge distillation. [35] proposed orthogonal weights modification (OWM) to enable networks to continually learn different mapping rules in a context-dependent way.

**Episodic memory-based approaches:** Augmenting the standard neural network with an external memory is a widely adopted practice [36, 37, 38]. In episodic memory based lifelong learning methods, a small reference memory is used for storing information from old tasks. GEM [1] and A-GEM [2] rotated the current gradient when the angle between the current gradient and the gradient computed on the reference memory is obtuse. MER [24] is a recently proposed lifelong learning algorithm which employed a meta-learning training strategy. In [39], a line of methods are proposed to select important samples to store in the memory in order to reduce memory size. Instead of storing samples, in [11] the authors proposed Orthogonal Gradient Descent (OGD) which projects the gradients on the new task onto a subspace in which the projected gradient will not affect the model's output on old tasks. [40] proposed *conceptor aided backprop* which is a variant of the back-propagation algorithm for avoiding catastrophic forgetting.

Our proposed schemes aim to improve episodic memory based approaches and are most related to [2]. Different from [2], the proposed schemes explicitly consider the performance of the model on old tasks and the new task in the process of rotating the current gradient. Our proposed schemes are also

related to several multi-task learning methods [41, 42, 43]. In [41, 42], the authors aimed at achieving a good balance between different tasks by learning to weigh the loss on each task . In contrast, our schemes directly leverage loss information in the context of lifelong learning for overcoming catastrophic forgetting. Compared with [43], instead of using the gradient norm information, our schemes and [1, 2] focus on rotating the direction of the current gradient. In [44], the authors consider gradient interference in multi-task learning while we focus on lifelong learning.

## 3  Lifelong Learning

Lifelong learning (LLL) [1, 2, 4, 23] considers the problem of learning a new task without degrading performance on old tasks, i.e., to avoid *catastrophic forgetting* [3, 4]. Suppose there are $T$ tasks which are characterized by $T$ datasets: $\{D_1, D_2, .., D_T\}$. Each dataset $D_t$ consists of a list of triplets $(x_i, y_i, t)$, where $y_i$ is the label of $i$-th example $x_i$, and $t$ is a task descriptor that indicates which task the example coming from. Similar to supervised learning, each dataset $D_t$ is split into a training set $D_t^{tr}$ and a test set $D_t^{te}$.

In the learning protocol introduced in [2], the tasks are separated into $D^{CV} = \{D_1, D_2, ..., D_{T^{CV}}\}$ and $D^{EV} = \{D_{T^{CV}+1}, D_{T^{CV}+2}, ..., D_T\}$. $D^{CV}$ is used for cross-validation to search for hyperparameters. $D^{EV}$ is used for actual training and evaluation. While searching for the hyperparameters, we can have multiple passes over the examples in $D^{CV}$, the training is performed on $D^{EV}$ with only a *single* pass over the examples [1, 2].

In lifelong learning, a given model $f(x; w)$ is trained sequentially on a series of tasks $\{D_{T^{CV}+1}, D_{T^{CV}+2}, ..., D_T\}$. When the model $f(x; w)$ is trained on task $D_t$, the goal is to predict the labels of the examples in $D_t^{te}$ by minimizing the empirical loss $\ell_t(w)$ on $D_t^{tr}$ in an online fashion without suffering accuracy drop on $\{D_{T^{CV}+1}^{te}, D_{T^{CV}+2}^{te}, ..., D_{t-1}^{te}\}$.

## 4  A Unified View of Episodic Memory Based Lifelong Learning

In this section, we provide a unified view for better understanding several episodic memory lifelong learning approach, including GEM [1] and A-GEM [2]. Due to space constraints, for the details of GEM and A-GEM, please refer to Appendix A.1.

GEM [1] and A-GEM [2] address the lifelong learning problem by utilizing a small episodic memory $M_k$ for storing a subset of the examples from task $k$. The episodic memory is populated by choosing examples uniformly at random for each task. While training on task $t$, the loss on the episodic memory $M_k$ can be computed as $\ell_{\text{ref}}(w_t; M_k) = \frac{1}{|M_k|} \sum_{(x_i, y_i) \in M_k} \ell(f(x_i; w_t), y_i)$, where $w_t$ is the weight of model during the training on task $t$.

In GEM and A-GEM, the lifelong learning model is trained via mini-batch stochastic gradient descent. We use $w_k^t$ to denote the weight when the model is being trained on the $k$-th mini-batch of task $t$. To establish the tradeoff between the performance on old tasks and the $t$-th task, we consider the following optimization problem with composite objective in each update step:

$$\min_w \alpha_1(w_k^t)\ell_t(w) + \alpha_2(w_k^t)\ell_{\text{ref}}(w) := \mathbb{E}_{\xi,\zeta}\left[\alpha_1(w_k^t)\ell_t(w; \xi) + \alpha_2(w_k^t)\ell_{\text{ref}}(w; \zeta)\right], \quad (1)$$

where $w \in \mathbb{R}^d$ is the parameter of the model, $\xi, \zeta$ are random variables with finite support, $\ell_t(w)$ is the expected training loss of the $t$-th task, $\ell_{\text{ref}}(w)$ is the expected loss calculated on the data stored in the episodic memory, $\alpha_1(w), \alpha_2(w) : \mathbb{R}^d \mapsto \mathbb{R}_+$ are real-valued mappings which control the relative importance of $\ell_t(w)$ and $\ell_{\text{ref}}(w)$ in each mini-batch.

Mathematically, we consider using the following update:

$$w_{k+1}^t = \arg\min_w \alpha_1(w_k^t) \cdot \ell_t(w; \xi) + \alpha_2(w_k^t) \cdot \ell_{\text{ref}}(w; \zeta). \quad (2)$$

The idea of GEM and A-GEM is to employ first-order methods (e.g., stochastic gradient descent) to approximately solve the optimization problem (2), where one-step stochastic gradient descent is performed with the initial point to be $w_k^t$:

$$w_{k+1}^t \leftarrow w_k^t - \eta\left(\alpha_1(w_k^t)\nabla\ell_t(w_k^t; \xi_k^t) + \alpha_2(w_k^t)\nabla\ell_{\text{ref}}(w_k^t; \zeta_k^t)\right), \quad (3)$$

where $\eta$ is the learning rate, $\xi_k^t$ and $\zeta_k^t$ are random variables with finite support, $\nabla\ell_t(w_k^t; \xi_k^t)$ and $\nabla\ell_{\text{ref}}(w_k^t; \zeta_k^t)$ are unbiased estimators of $\nabla\ell_t(w_k^t)$ and $\nabla\ell_{\text{ref}}(w_k^t)$ respectively. The quantity $\alpha_1(w_k^t)\nabla\ell_t(w_k^t; \xi_k^t) + \alpha_2(w_k^t)\nabla\ell_{\text{ref}}(w_k^t; \zeta_k^t)$ is referred to as the *mixed stochastic gradient*.

**Algorithm 1** The proposed improved schemes for episodic memory based lifelong learning.

---
1: $M \leftarrow \{\}$
2: **for** $t \leftarrow 1$ to $T$ **do**
3:     **for** $k \leftarrow 1$ to $|D_t^{tr}|$ **do**
4:         **if** $M \neq \{\}$ **then**
5:             $\zeta_k^t \leftarrow \text{SAMPLE}(M)$
6:             MEGA-I: choose $\alpha_1$ and $\alpha_2$ based on Equation 6.
7:             MEGA-II: choose $\alpha_1$ and $\alpha_2$ as in Appendix A.3.
8:         **else**
9:             Set $\alpha_1(w) = 1$ and $\alpha_2(w) = 0$.
10:         **end if**
11:         Update $w_k^t$ using Eq. 3.
12:         $M \leftarrow M \bigcup (\xi_k^t, y_k^t)$
13:         Discard the samples added initially if $M$ is full.
14:     **end for**
15: **end for**

---

In A-GEM, $\nabla \ell_{\text{ref}}(w_k^t; \xi_k^t)$ is the reference gradient computed based on a random subset from the episodic memory $M$ of all past tasks, where $M = \cup_{k<t} M_k$. And $\alpha_1(w_k^t)$ and $\alpha_2(w_k^t)$ can be written as,

$$\alpha_1(w_k^t) = 1, \alpha_2(w_k^t) = \mathbf{I}_{\langle \nabla \ell_{\text{ref}}(w_k^t; \zeta_k^t), \nabla \ell_t(w_k^t; \xi_k^t) \rangle \leq 0} \times \left( -\frac{\nabla \ell_t(w_k^t; \xi_k^t)^\top \nabla \ell_{\text{ref}}(w_k^t; \zeta_k^t)}{\nabla \ell_{\text{ref}}(w_k^t; \zeta_k^t)^\top \nabla \ell_{\text{ref}}(w_k^t; \zeta_k^t)} \right), \quad (4)$$

where $\mathbf{I}_u$ is the indicator function, which is 1 if $u$ holds and otherwise 0.

In GEM, there are $t-1$ reference gradients based on the previous $t-1$ tasks respectively. In this case, $\nabla \ell_{\text{ref}}(w_k^t; \zeta_k^t) = [g_1, \ldots, g_{t-1}] \in \mathbb{R}^{d \times (t-1)}$ and $\alpha_2(w_k^t) \in \mathbb{R}^{t-1}$, where $g_1, \ldots, g_{t-1}$ are reference gradients based on $M_1, \ldots, M_{t-1}$ respectively. In GEM,

$$\alpha_1(w_k^t) = 1, \alpha_2(w_k^t) = v_*, \quad (5)$$

where $v_*$ is the optimal solution for the quadratic programming problem (11) in Appendix A.1.

As we can see from the formulation (4) and (5), both A-GEM and GEM set $\alpha_1(w) = 1$ in the whole training process. It means that both A-GEM and GEM always put the same emphasis on the current task, regardless of how the loss changes over time. During the lifelong learning process, the current loss and the reference loss are changing dynamically in each mini-batchs, and consistently choosing $\alpha_1(w) = 1$ may not capture a good balance between current loss and the reference loss.

## 5 Mixed Stochastic Gradient

In this section, we introduce Mixed Stochastic Gradient (MEGA) to address the limitations of GEM and A-GEM. We adopt the way of A-GEM for computing the reference loss due to the better performance of A-GEM over GEM. Instead of consistently putting the same emphasis on the current task, the proposed schemes allow adaptive balancing between current task and old tasks. Specifically, MEGA-I and MEGA-II utilize the loss information during training which is ignored by GEM and A-GEM. In Section 5.1, we propose MEGA-I which utilizes loss information to balance the reference gradient and the current gradient. In Section 5.2, we propose MEGA-II which considers the cosine similarities between the update direction with the current gradient and the reference gradient.

### 5.1 MEGA-I

We introduce MEGA-I which is an adaptive loss-based approach to balance the current task and old tasks by only leveraging loss information. We introduce a pre-defined sensitivity parameter $\epsilon$ similar to [45]. In the update of (3), we set

$$\begin{cases} \alpha_1(w) = 1, \alpha_2(w) = \ell_{\text{ref}}(w; \zeta)/\ell_t(w; \xi) & \text{if } \ell_t(w; \xi) > \epsilon \\ \alpha_1(w) = 0, \alpha_2(w) = 1 & \text{if } \ell_t(w; \xi) \leq \epsilon, \end{cases} \quad (6)$$

Intuitively, if $\ell_t(w; \xi)$ is small, then the model performs well on the current task and MEGA-I focuses on improving performance on the data stored in the episodic memory. To this end, MEGA-I chooses

$\alpha_1(w) = 0$, $\alpha_2(w) = 1$. Otherwise, when the current loss is larger than $\epsilon$, MEGA-I keeps the balance of the two terms of mixed stochastic gradient according to $\ell_t(w; \xi)$ and $\ell_{ref}(w; \zeta)$. Intuitively, if $\ell_t(w; \xi)$ is relatively larger than $\ell_{ref}(w; \zeta)$, then MEGA-I puts less emphasis on the reference gradient and vice versa.

Compared with GEM and A-GEM update rule in (5) and (4), MEGA-I makes an improvement to handle the case of overfitting on the current task (i.e., $\ell_t(w; \xi) \leq \epsilon$), and to dynamically change the relative importance of the current and reference gradient according to the losses on the current task and previous tasks.

### 5.2 MEGA-II

The magnitude of MEGA-I's mixed stochastic gradient depends on the magnitude of the current gradient and the reference gradient, as well as the losses on the current task and the episodic memory. Inspired by A-GEM, MEGA-II's mixed gradient is obtained from a rotation of the current gradient whose magnitude only depends on the current gradient.

The key idea of the MEGA-II is to first appropriately rotate the stochastic gradient calculated on the current task (i.e., $\nabla \ell_t(w_k^t; \xi_k^t)$) by an angle $\theta_k^t$, and then use the rotated vector as the mixed stochastic gradient to conduct the update (3) in each mini-batch. For simplicity, we omit the subscript $k$ and superscript $t$ later on unless specified.

We use $\mathbf{g}_{mix}$ to denote the desired mixed stochastic gradient which has the same magnitude as $\nabla \ell_t(w; \xi)$. Specifically, we look for the mixed stochastic gradient $\mathbf{g}_{mix}$ which direction aligns well with both $\nabla \ell_t(w; \xi)$ and $\nabla \ell_{ref}(w; \zeta)$. Similar to MEGA-I, we use the loss-balancing scheme and desire to maximize

$$\ell_t(w; \xi) \cdot \frac{\langle \mathbf{g}_{mix}, \nabla \ell_t(w; \xi) \rangle}{\|\mathbf{g}_{mix}\|_2 \cdot \|\nabla \ell_t(w; \xi)\|_2} + \ell_{ref}(w; \zeta) \cdot \frac{\langle \mathbf{g}_{mix}, \nabla \ell_{ref}(w; \zeta) \rangle}{\|\mathbf{g}_{mix}\|_2 \cdot \|\nabla \ell_{ref}(w; \zeta)\|_2}, \quad (7)$$

which is equivalent to find an angle $\theta$ such that

$$\theta = \arg \max_{\beta \in [0,\pi]} \ell_t(w; \xi) \cos(\beta) + \ell_{ref}(w; \zeta) \cos(\tilde{\theta} - \beta). \quad (8)$$

where $\tilde{\theta} \in [0, \pi]$ is the angle between $\nabla \ell_t(w; \xi)$ and $\nabla \ell_{ref}(w; \zeta)$, and $\beta \in [0, \pi]$ is the angle between $\mathbf{g}_{mix}$ and $\nabla \ell_t(w; \xi)$. The closed form of $\theta$ is $\theta = \frac{\pi}{2} - \alpha$, where $\alpha = \arctan\left(\frac{k + \cos \tilde{\theta}}{\sin \tilde{\theta}}\right)$ and $k = \ell_t(w; \xi)/\ell_{ref}(w; \zeta)$ if $\ell_{ref}(w; \zeta) \neq 0$ otherwise $k = +\infty$. The detailed derivation of the closed form of $\theta$ can be found in Appendix A.2. Here we give some discussions of several special cases of Eq. (8).

- When $\ell_{ref}(w; \zeta) = 0$, then $\theta = 0$, and in this case $\alpha_1(w) = 1$, $\alpha_2(w) = 0$ in (3), the mixed stochastic gradient reduces to $\nabla \ell_t(w; \xi)$. In the lifelong learning setting, $\ell_{ref}(w; \zeta) = 0$ implies that there is almost no catastrophic forgetting, and hence we can update the model parameters exclusively for the current task by moving in the direction of $\nabla \ell_t(w; \xi)$.
- When $\ell_t(w; \xi) = 0$, then $\theta = \tilde{\theta}$, and in this case $\alpha_1(w) = 0$, $\alpha_2(w) = \|\nabla \ell_t(w; \xi)\|_2 / \|\nabla \ell_{ref}(w; \zeta)\|_2$, provided that $\|\nabla \ell_{ref}(w; \zeta)\|_2 \neq 0$ (define 0/0=0). In this case, the direction of the mixed stochastic gradient is the same as the stochastic gradient calculated on the data in the episodic memory (i.e., $\ell_{ref}(w; \zeta)$). In the lifelong learning setting, this update can help improve the performance on old tasks, i.e., avoid catastrophic forgetting.

After we find the desired angle $\theta$, we can calculate $\alpha_1(w)$ and $\alpha_2(w)$ in Eq. (3), as shown in Appendix A.3. It is worth noting that different from GEM and A-GEM which always set $\alpha_1(w) = 1$, the proposed MEGA-I and MEGA-II adaptively adjust $\alpha_1$ and $\alpha_2$ based on performance of the model on the current task and the episodic memory. Please see Algorithm 1 for the summary of the algorithm.

## 6 Experiments

### 6.1 Experimental Settings and Evaluation Protocol

We conduct experiments on commonly used lifelong learning bencnmarks: **Permutated MNIST** [4], **Split CIFAR** [21], **Split CUB** [2], **Split AWA** [2]. The details of the datasets can be found in

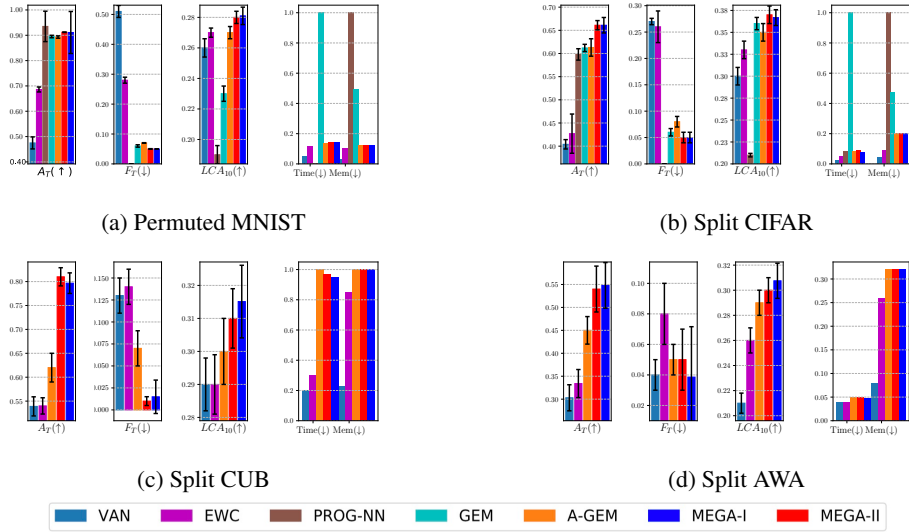

(a) Permuted MNIST       (b) Split CIFAR

(c) Split CUB       (d) Split AWA

VAN   EWC   PROG-NN   GEM   A-GEM   MEGA-I   MEGA-II

Figure 1: Performance of lifelong learning models across different measures on **Permuted MNIST**, **Split CIFAR**, **Split CUB** and **Split AWA**.

Appendix A.4. We compare MEGA-I and MEGA-II with several baselines, including VAN [2], MULTI-TASK [2], EWC [4], PI [21], MAS [29], RWALK [22], ICARL [46], PROG-NN [23], MER [24], GEM [1] and A-GEM [2]. In particular, in VAN [2], a single network is trained continuously on a sequence of tasks in a standard supervised learning manner. In MULTI-TASK [2], a single network is trained on the shuffled data from all the tasks with a single pass.

To be consistent with the previous works [1, 2], for **Permuted MNIST** we adopt a standard fully-connected network with two hidden layers. Each layer has 256 units with ReLU activation. For **Split CIFAR** we use a reduced ResNet18. For **Split CUB** and **Split AWA**, we use a standard ResNet18 [47]. We use Average Accuracy ($A_T$), Forgetting Measure ($F_T$) and Learning Curve Area (LCA) [1, 2] for evaluating the performance of lifelong learning algorithms. $A_T$ is the average test accuracy and $F_T$ is the degree of accuracy drop on old tasks after the model is trained on all the $T$ tasks. LCA is used to assess the learning speed of different lifelong learning algorithms. Please see Appendix A.5 for the definitions of different metrics.

To be consistent with [2], for episodic memory based approaches, the episodic memory size for each task is 250, 65, 50, and 100, and the batch size for computing the gradients on the episodic memory (if needed) is 256, 256, 128 and 128 for MNIST, CIFAR, CUB and AWA, respectively. To fill the episodic memory, the examples are chosen uniformly at random for each task as in [2]. For each dataset, 17 tasks are used for training and 3 tasks are used for hyperparameter search. For the baselines, we use the best hyperparameters found by [2]. For the detailed hyperparameters, please see Appendix G of [2]. For MER [24], we reuse the best hyperparameters found in [24]. In MEGA-I, the $\epsilon$ is chosen from $\{10^{-5:1:-1}\}$ via the 3 validation tasks. For MEGA-II, we reuse the hyperparameters from A-GEM [2]. All the experiments are done on 8 NVIDIA TITAN RTX GPUs. The code can be found in the supplementary material.

## 6.2 Results

### 6.2.1 MEGA VS Baselines

In Fig. 1 we show the results across different measures on all the benchmark datasets. We have the following observations. First, MEGA-I and MEGA-II outperform all baselines across the benchmarks, except that PROG-NN achieves a slightly higher accuracy on **Permuted MNIST**. As we can see from the memory comparison, PROG-NN is very memory inefficient since it allocates a new network for each task, thus the number of parameters grows super-linearly with the number of tasks. This becomes problematic when large networks are being used. For example, PROG-NN runs out of memory on **Split CUB** and **Split AWA** which prevents it from scaling up to real-life problems. On other datasets, MEGA-I and MEGA-II consistently perform better than all the baselines. From

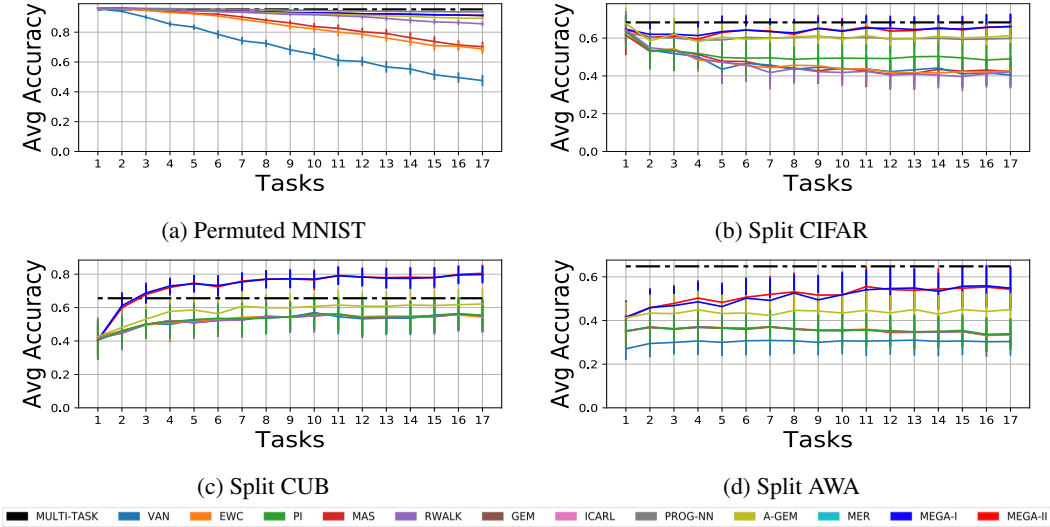

(a) Permuted MNIST            (b) Split CIFAR

(c) Split CUB            (d) Split AWA

Figure 2: Evolution of average accuracy during the lifelong learning process.

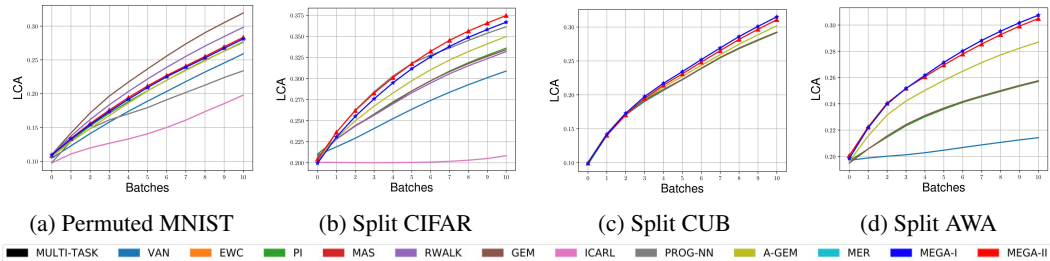

(a) Permuted MNIST    (b) Split CIFAR    (c) Split CUB    (d) Split AWA

Figure 3: LCA of first ten mini-batches on different datasets.

Fig. 2 we can see that on **Split CUB**, MEGA-I and MEGA-II even surpass the multi-task baseline which is previously believed as an upper bound performance of lifelong learning algorithms [2]. Second, MEGA-I and MEGA-II achieve the lowest Forgetting Measure across all the datasets which indicates their ability to overcome catastrophic forgetting. Third, MEGA-I and MEGA-II also obtain a high LCA across all the datasets which shows that MEGA-I and MEGA-II also learn quickly. The evolution of LCA in the first ten mini-batches across all the datasets is shown in Fig. 3. Last, we can observe that MEGA-I and MEGA-II achieve similar results in Fig. 1. For detailed results, please refer to Table 2 and Table 3 in Appendix A.6.

In Fig. 2 we show the evolution of average accuracy during the lifelong learning process. As more tasks are added, while the average accuracy of the baselines generally drops due to catastrophic forgetting, MEGA-I and MEGA-II can maintain and even improve their performance. In the next section, we will show that MEGA-II outperforms MEGA-I when the number of examples is limited per task.

### 6.2.2 MEGA-II Outperforms Other Baselines and MEGA-I When the Number of Examples is Limited

Inspired by few-shot learning [48, 49, 50, 51], in this section we consider a more challenging setting for lifelong learning where each task only has a limited number of examples.

We construct 20 tasks with $X$ number of examples per task, where $X$ = 200, 400 and 600. The way to generate the tasks is the same as in **Permuted MNIST**, that is, a fixed random permutation of input pixels is applied to all the examples for a particular task. The running time is measured on one NVIDIA TITAN RTX GPU. The results of average accuracy are shown in Fig. 4(a). We can see that MEGA-II outperforms all the baseline and MEGA-I when the number of examples is limited. In Fig. 4(b), we show the execution time for each method, the proposed MEGA-I and MEGA-II are

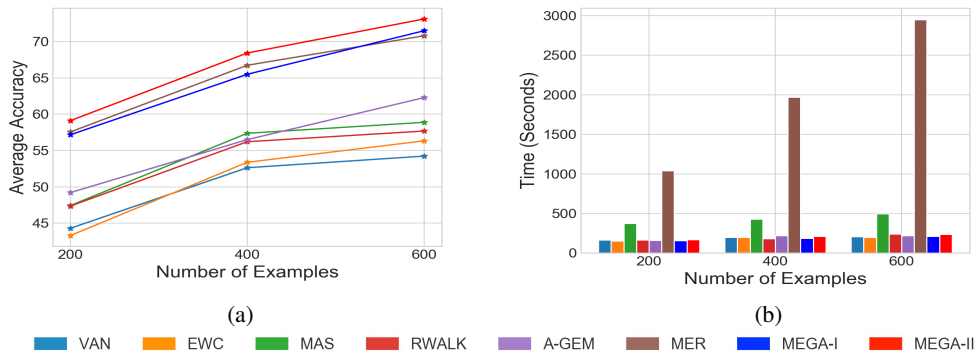

Figure 4: The average accuracy and execution time when the number of examples is limited.

computational efficient compared with other methods. Compared with MER [24] which achieves similar results to MEGA-I and MEGA-II, MEGA-I and MEGA-II is much more time efficient since it does not rely on the meta-learning procedure.

We analyze the reason why MEGA-II outperforms MEGA-I when the number of examples is small. In this case, it is difficult to learn well on the current task, so the magnitude of the current loss and the current gradient's norm are both large. MEGA-I directly balances the reference gradient and the current gradient, and the mixed stochastic gradient is dominated by the current gradient and it suffers from catastrophic forgetting. In contrast, MEGA-II balances the cosine similarity between gradients. Even if the norm of the current gradient is large, MEGA-II still allows adequate rotation of the direction of the current gradient to be closer to that of the reference gradient to alleviate catastrophic forgetting. We validate our claims by detailed analysis, which can be found in Appendix A.7.

### 6.2.3 Ablation Studies

In this section, we include detailed ablation studies to analyze the reason why the proposed schemes can improve current episodic memory based lifelong learning methods. For MEGA-I, we consider the setting that both $\alpha_1(w) = 1$ and $\alpha_2(w) = 1$ during the training process. This ablation study is to show the effectiveness of the adaptive loss balancing scheme. For MEGA-II, we consider the setting that $\ell_t = \ell_{\text{ref}}$ in Eq. 7 to verify the effectiveness of the proposed gradient rotation scheme over A-GEM. The experimental settings are the same as Section 6.1. The results are shown in Table 1.

Table 1: Comparison of MEGA-I, MEGA-I ($\alpha_1(w) = 1, \alpha_2(w) = 1$), MEGA-II, MEGA-II ($\ell_t = \ell_{\text{ref}}$) and A-GEM.

| Method | Permuted MNIST $A_T$ (%) | Split CIFAR $A_T$(%) | Split CUB $A_T$(%) | Split AWA $A_T$(%) |
|---|---|---|---|---|
| MEGA-I | $91.10 \pm 0.08$ | $66.10 \pm 1.67$ | $79.67 \pm 2.15$ | $\mathbf{54.82 \pm 4.97}$ |
| MEGA-I ($\alpha_1(w) = 1, \alpha_2(w) = 1$) | $90.66 \pm 0.09$ | $64.65 \pm 1.98$ | $79.44 \pm 2.98$ | $53.60 \pm 5.21$ |
| MEGA-II | $\mathbf{91.21 \pm 0.10}$ | $\mathbf{66.12 \pm 1.93}$ | $\mathbf{80.58 \pm 1.94}$ | $54.28 \pm 4.84$ |
| MEGA-II ($\ell_t = \ell_{\text{ref}}$) | $91.15 \pm 0.12$ | $58.04 \pm 1.89$ | $68.60 \pm 1.98$ | $47.95 \pm 4.54$ |
| A-GEM | $89.32 \pm 0.46$ | $61.28 \pm 1.88$ | $61.82 \pm 3.72$ | $44.95 \pm 2.97$ |

In Table 1, we observe that MEGA-I achieves higher average accuracy than MEGA-I ($\alpha_1(w) = 1, \alpha_2(w) = 1$) by considering an adaptive loss balancing scheme. We also see that except on **Split CIFAR**, MEGA-II ($\ell_t = \ell_{\text{ref}}$) outperforms A-GEM on all the datasets. This demonstrates the benefits of the proposed approach for rotating the current gradient. By considering the loss information as in MEGA-II, we further improve the results on all the datasets. This shows that both of the components (the rotation of the current gradient and loss balancing) contribute to the improvements of the proposed schemes.

## 7 Conclusion

In this paper, we cast the lifelong learning problem as an optimization problem with composite objective, which provides a unified view to cover current episodic memory based lifelong learning algorithms. Based on the unified view, we propose two improved schemes called MEGA-I and

MEGA-II. Extensive experimental results show that the proposed MEGA-I and MEGA-II achieve superior performance, significantly advancing the state-of-the-art on several standard benchmarks.

## Acknowledgment

This work was supported in part by CRISP, one of six centers in JUMP, an SRC program sponsored by DARPA. This work is also supported by NSF CHASE-CI #1730158, NSF FET #1911095, NSF CC* NPEO #1826967, NSF #1933212, NSF CAREER Award #1844403. The paper was also funded in part by SRC AIHW grants.

## Broader Impact

In this paper, researchers introduce a unified view on current episodic memory based lifelong learning methods and propose two improved schemes: MEGA-I and MEGA-II. The proposed schemes demonstrate superior performance and advance the state-of-the-art on several lifelong learning benchmarks.

The unified view embodies existing episodic memory based lifelong learning methods in the same general framework. The proposed MEGA-I and MEGA-II significantly improve existing episodic memory based lifelong learning such as GEM [1] and A-GEM [2]. The proposed schemes enable machine learning models to acquire the ability to learn tasks sequentially without *catastrophic forgetting*. Machine learning models with continual learning capability can be applied in image classification [1] and natural language processing [52].

The proposed lifelong learning algorithms can be applied in several real-world applications such as on-line advertisement, fraud detection, climate change monitoring, recommendation systems, industrial manufacturing and so on. In all these applications, the data are arriving sequentially and the data distribution may change over time. For example, in recommendation systems, the users' preferences may vary due to their aging, personal financial status or health condition. The machine learning models without continual learning capability may not capture such dynamics. The proposed lifelong learning schemes are able to address this issue.

The related applications have a broad range of societal implications: the use of lifelong recommendation systems can bring several benefits such as reducing the cost of model retraining and providing better user experience. However, such systems may have the concerns of data privacy. Lifelong recommendation systems can increase customer satisfaction. In the mean time, this system needs to store part of user data which may compromise user's privacy.

Our proposed lifelong learning schemes also are closely related to other machine learning research areas, including multi-task learning, transfer learning, federated learning, few-shot learning and so on. In transfer learning, when the source domain and the target domain are different, it is crucial to develop techniques that can reduce the *negative transfer* between the domains during the fine-tuning process. We expect that our proposed approaches can be leveraged to resolve the issue of negative transfer.

We encourage researchers to further investigate the merits and shortcomings of our proposed methods. In particular, we recommend researchers and policymakers to look into lifelong learning systems without storing examples from past tasks. Such systems do not jeopardize users' privacy and can be deployed in critical scenarios such as financial applications.

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
