[Supplementary Material · 4462_Supplementary.pdf]

# A Appendix

## A.1 Gradient Episodic Memory (GEM) and Averaged Gradient Episodic Memory (A-GEM)

GEM ensures that each update on the $t$-th task will not increase the loss on the episodic memory, that is,

$$\text{minimize}_w \ell(w; D_t^{tr}) \quad \text{s.t.} \quad \ell(w; M_k) \leq \ell(w_{t-1}; M_k) \quad \forall k < t \tag{9}$$

To inspect the increase of loss on the episodic memory, GEM computes the gradient $g$ on the current task and the reference gradient $g_k$ on the episodic memory $M_k$. When the angle between $g$ and $g_k$ is obtuse, GEM projects the current gradient $g$ to have a right or acute angle with $g_k$,

$$\text{minimize}_{g_{\text{true}}} \frac{1}{2} \|g - g_{\text{true}}\|_2^2 \quad \text{s.t.} \quad g_{\text{true}}^\top g_k \geq 0 \quad \forall k < t \tag{10}$$

GEM solves above optimization problem via quadratic programming in the dual space with $v \in \mathbb{R}^{(t-1) \times 1}$:

$$\text{minimize}_v v^\top G^\top G v + g^\top G v \quad \text{s.t.} \quad v \geq 0 \tag{11}$$

where $G = -(g_1, ..., g_{t-1}) \in \mathbb{R}^{d \times (t-1)}$, $g \in \mathbb{R}^{d \times 1}$, and $d$ is the number of parameters in the neural network. After obtaining the solution $v^*$, the gradient used for updating the model can be computed as $g_{true} = Gv^* + g$.

A-GEM [2] improves the efficiency of GEM by preventing the average episodic memory loss from increasing. In A-GEM, $G$ is replaced by $-g_{ref}$ which is the gradient computed on a random subset of the examples from all old tasks. And $v^*$ is replaced with a single scalar which can be computed in closed form as $\frac{g^\top g_{\text{ref}}}{g_{\text{ref}}^\top g_{\text{ref}}}$.

## A.2 Closed-Form Solution of $\theta$

Define $k = \ell_t / \ell_{\text{ref}}$, then we have

$$\begin{aligned}
\theta &= \arg\max_\beta \left[ k\cos(\beta) + \cos(\tilde{\theta} - \beta) \right] \\
&= \arg\max_\beta \left[ k\cos\beta + \cos\tilde{\theta}\cos\beta + \sin\tilde{\theta}\sin(\beta) \right] \\
&= \arg\max_\beta \left[ \cos\beta \left( k + \cos\tilde{\theta} \right) + \sin\tilde{\theta}\sin(\beta) \right] \\
&= \arg\max_\beta \left[ \left( \frac{k + \cos\tilde{\theta}}{\sin\tilde{\theta}} \cos\beta + \sin\beta \right) \cdot \sin\tilde{\theta} \right] \\
&= \arg\max_\beta \left[ \frac{\sin\tilde{\theta}}{\cos\alpha} \left( \sin\alpha\cos\beta + \cos\alpha\sin\beta \right) \right] \\
&= \arg\max_\beta \left[ \frac{\sin\tilde{\theta}}{\cos\alpha} \sin(\alpha + \beta) \right] \\
&= \frac{\pi}{2} - \alpha
\end{aligned} \tag{12}$$

, where $\alpha = \arctan\left( \frac{k + \cos\tilde{\theta}}{\sin\tilde{\theta}} \right)$.

 **A.3 Some Derivations**

488 For notation simplicity, we use $\mathbf{g}$, $\hat{\mathbf{g}}$, $a$, $b$ to replace $\nabla\ell_t(w;\xi)$, $\nabla\ell_{\text{ref}}(w;\zeta)$, $\alpha_1(w)$, $\alpha_2(w)$ respec-
489 tively. If $\mathbf{g} = \hat{\mathbf{g}}$, then $a = 1$, $b = 0$. Otherwise, the goal is to solve

$$
\begin{aligned}
a\mathbf{g}^\top\mathbf{g} + b\mathbf{g}^\top\hat{\mathbf{g}} &= \|\mathbf{g}\|_2^2\cos\theta \\
a\mathbf{g}^\top\hat{\mathbf{g}} + b\|\hat{\mathbf{g}}\|_2^2 &= \|\mathbf{g}\|\|\hat{\mathbf{g}}\|\cos(\tilde{\theta} - \theta)
\end{aligned}
\tag{13}
$$

490 The solution of (13) is

$$
\begin{aligned}
a &= \frac{1}{\|\mathbf{g}\|_2^2\|\hat{\mathbf{g}}\|_2^2 - \mathbf{g}^\top\hat{\mathbf{g}}}\left[\|\hat{\mathbf{g}}\|_2^2\|\mathbf{g}\|_2^2\cos\theta - (\mathbf{g}^\top\hat{\mathbf{g}})\|\mathbf{g}\|_2\|\hat{\mathbf{g}}\|_2\cos(\tilde{\theta} - \theta)\right] \\
b &= \frac{1}{\|\mathbf{g}\|_2^2\|\hat{\mathbf{g}}\|_2^2 - \mathbf{g}^\top\hat{\mathbf{g}}}\left[-(\mathbf{g}^\top\hat{\mathbf{g}})\|\mathbf{g}\|_2^2\cos\theta + \|\mathbf{g}\|_2^3\|\hat{\mathbf{g}}\|_2\cos(\tilde{\theta} - \theta)\right]
\end{aligned}
\tag{14}
$$

491 **A.4 Datasets**

492 In the experiments, we consider the following four conventional lifelong learning benchmarks,

493 • **Permuted MNIST** [4]: this is a variant of standard MNIST dataset [49] of handwritten
494 digits with 20 tasks. Each task has a fixed random permutation of the input pixels which is
495 applied to all the images of that task.

496 • **Split CIFAR** [21]: this dataset consists of 20 disjoint subsets of CIFAR-100 dataset [50],
497 where each subset is formed by randomly sampling 5 classes without replacement from the
498 original 100 classes.

499 • **Split CUB** [2]: the CUB dataset [51] is split into 20 disjoint subsets by randomly sampling
500 10 classes without replacement from the original 200 classes.

501 • **Split AWA** [2]: this dataset consists of 20 subsets of the AWA dataset [52]. Each subset is
502 constructed by sampling 5 classes with replacement from a total of 50 classes. Note that the
503 same class can appear in different subsets. As in [2], in order to guarantee that each training
504 example only appears once in the learning process, based on the occurrences in different
505 subsets the training data of each class is split into disjoint sets. In the learning process, the
506 weights of the classifier of each class are randomly initialized within each head without any
507 transfer from the previous occurrence of the class in past tasks.

508 **A.5 Evaluation Metrics**

509 Average Accuracy and Forgetting Measure [22] are common used metrics for evaluating performance
510 of lifelong learning algorithms. In [2], the authors introduce another metric, called Learning Curve
511 Area (LCA), to assess the learning speed of different lifelong learning algorithms.

512 Suppose there are $N_k$ mini-batches in the training set of task $D_k$. Similar to [2], we define $a_{k,i,j}$
513 as the accuracy on the test set of task $D_j$ after the model is trained on the $i$-th mini-batch of task
514 $D_k$. Generally, suppose the model $f(x;w)$ is trained on a sequence of $T$ tasks $\{D_1, D_2, ..., D_T\}$.
515 Average Accuracy and Forgetting Measure after the model is trained on the task $D_k$ are defined as

$$
A_k = \frac{1}{k}\sum_{j=1}^{k} a_{k,M_k,j} \quad F_k = \frac{1}{k-1}\sum_{j=1}^{k-1} f_j^k,
\tag{15}
$$

516 where $f_j^k = \max_{l\in\{1,2,..,k-1\}} a_{l,M_l,j} - a_{k,M_k,j}$. Clearly, $A_T$ is the average test accuracy and $F_T$
517 assesses the degree of accuracy drop on old tasks after the model is trained on all the $T$ tasks. Learning
518 Curve Area (LCA) [2] at $\beta$ is defined as,

$$
\text{LCA}_\beta = \frac{1}{\beta + 1}\sum_{b=0}^{\beta} Z_b,
\tag{16}
$$

519 where $Z_b = \frac{1}{T}\sum_{k=1}^{T} a_{k,b,k}$. Intuitively, LCA measures the learning speed of different lifelong
520 learning algorithms. A higher value of LCA indicates that the model learns quickly. We refer the
521 readers to [2] for more details about LCA.

**A.6 RESULT TABLES**

523 In Table 2 and Table 3 we show the detailed results of all the methods on different benchmarks.

Table 2: The results of Average Accuracy ($A_T$), Forgetting Measure ($F_T$) and LCA of different methods on **Permuted MNIST** and **Split CIFAR**. The results are averaged across 5 runs with different random seeds.

| Methods | Permuted MNIST | | | Split CIFAR | | |
|---|---|---|---|---|---|---|
| | $A_T(\%)$ | $F_T$ | $LCA_{10}$ | $A_T(\%)$ | $F_T$ | $LCA_{10}$ |
| VAN | 47.55±2.37 | 0.52±0.026 | 0.259±0.005 | 40.44±1.02 | 0.27±0.006 | 0.309±0.011 |
| EWC | 68.68±0.98 | 0.28±0.010 | 0.276±0.002 | 42.67±4.24 | 0.26±0.039 | 0.336±0.010 |
| MAS | 70.30±1.67 | 0.26±0.018 | 0.298±0.006 | 42.35±3.52 | 0.26±0.030 | 0.332±0.010 |
| RWALK | 85.60±0.71 | 0.08±0.007 | **0.319**±0.003 | 42.11±3.69 | 0.27±0.032 | 0.334±0.012 |
| MER | - | - | - | 37.27±1.68 | 0.03±0.030 | 0.051±0.101 |
| PROG-NN | **93.55**±0.06 | **0.0**±0.000 | 0.198±0.006 | 59.79±1.23 | **0.0**±0.000 | 0.208±0.002 |
| GEM | 89.50±0.48 | 0.06±0.004 | 0.230±0.005 | 61.20±0.78 | 0.06±0.007 | 0.360±0.007 |
| A-GEM | 89.32±0.46 | 0.07±0.004 | 0.277±0.008 | 61.28±1.88 | 0.09±0.018 | 0.350±0.013 |
| MEGA-I | 91.10±0.08 | 0.05±0.001 | 0.281± 0.005 | 66.10±1.67 | 0.05±0.014 | 0.366±0.009 |
| MEGA-II | 91.21±0.10 | 0.05±0.001 | 0.283±0.004 | **66.12**±1.94 | 0.06±0.015 | **0.375**±0.012 |

Table 3: The results of Average Accuracy ($A_T$), Forgetting Measure ($F_T$) and LCA of different methods on **Split CUB** and **Split AWA**. The results are averaged across 10 runs with different random seeds.

| Methods | Split CUB | | | Split AWA | | |
|---|---|---|---|---|---|---|
| | $A_T(\%)$ | $F_T$ | $LCA_{10}$ | $A_T(\%)$ | $F_T$ | $LCA_{10}$ |
| VAN | 53.89±2.00 | 0.13±0.020 | 0.292±0.008 | 30.35±2.81 | **0.04**±0.013 | 0.214±0.008 |
| EWC | 53.56±1.67 | 0.14±0.024 | 0.292±0.009 | 33.43±3.07 | 0.08±0.021 | 0.257±0.011 |
| MAS | 54.12±1.72 | 0.13±0.013 | 0.293±0.008 | 33.83±2.99 | 0.08±0.022 | 0.257±0.011 |
| RWALK | 54.11±1.71 | 0.13±0.013 | 0.293±0.009 | 33.63±2.64 | 0.08±0.023 | 0.258±0.011 |
| PI | 55.04±3.05 | 0.12±0.026 | 0.292±0.010 | 33.86±2.77 | 0.08±0.022 | 0.259±0.011 |
| A-GEM | 61.82±3.72 | 0.08±0.021 | 0.302±0.011 | 44.95±2.97 | 0.05±0.014 | 0.287±0.012 |
| MEGA-I | 79.67±2.15 | 0.01±0.019 | **0.315**±0.011 | **54.82**±4.97 | 0.04±0.034 | **0.307**±0.014 |
| MEGA-II | **80.58**±1.94 | **0.01**±0.017 | 0.311±0.010 | 54.28±4.84 | 0.05±0.040 | 0.305±0.015 |

524 **A.7 Detailed Analysis of MEGA-I and MEGA-II**

525 In this section, we present a detailed analysis on the reason that why the MEGA-II outperforms
526 MEGA-I significantly when the number of examples is limited. Define $k_1 = \frac{\ell_t}{\ell_{\text{ref}}}$, $k_2 = \frac{\|\nabla\ell_t(w;\xi)\|}{\|\nabla\ell_{\text{ref}}(w;\zeta)\|}$.
527 We denote the angles between the mixed gradient $\mathbf{g}_{\text{mix}}$ and the current gradient $\nabla\ell_t(w;\xi)$ calculated
528 by MEGA-I and MEGA-II by $\theta_1$ and $\theta_2$ respectively. In Appendix A.2, we know that

$$\cos\theta_2 = \frac{k_1 + \cos\tilde{\theta}}{\sqrt{k_1^2 + 2k_1\cos\tilde{\theta} + 1}}. \tag{17}$$

529 Now we derive the closed form of $\cos\theta_1$. For simplicity, we only consider the case where $\ell_t(w;\xi) \geq \epsilon$.
530 By formula (6), we know that $\mathbf{g}_{\text{mix}} = \nabla\ell_t(w;\xi) + \frac{\ell_{\text{ref}}}{\ell_t}\nabla\ell_{\text{ref}}(w;\zeta)$. Define $\mathbf{g}_t = \ell_t(w;\xi)$, $\mathbf{g}_{\text{ref}} = $
531 $\nabla\ell_{\text{ref}}(w;\zeta)$. By some algebra, we can show that

$$\mathbf{g}_{\text{mix}} = \frac{\ell_t}{\ell_{\text{ref}}}\|\mathbf{g}_{\text{ref}}\|\left(k_1 k_2 \frac{\mathbf{g}_t}{\|\mathbf{g}_t\|} + \frac{\mathbf{g}_{\text{ref}}}{\|\mathbf{g}_{\text{ref}}\|}\right).$$

532 Hence, we have

$$\cos\theta_1 = \frac{\mathbf{g}_{\text{mix}}^\top\mathbf{g}_t}{\|\mathbf{g}_{\text{mix}}\|\|\mathbf{g}_t\|} = \frac{k_1 k_2 + \cos\tilde{\theta}}{\sqrt{k_1^2 k_2^2 + 2k_1 k_2\cos\tilde{\theta} + 1}}. \tag{18}$$

533 Comparing (18) and (17), and noting that the function $f(k) = \frac{k+\cos\tilde{\theta}}{\sqrt{k^2+2k\cos\tilde{\theta}+1}}$ is a monotonically
534 increasing function with respect to $k$ for $k \geq 0$, we know that if $k_1 k_2 \geq k_1$, i.e., $k_2 \geq 1$, then
535 $\cos\theta_1 \geq \cos\theta_2$, which means $\theta_1 \leq \theta_2$.

(a) 200 Examples      (b) 600 Examples      (c) 55000 Examples

Figure 5: Count versus $\log(k_2)$, where $k_2 = \frac{\|\mathbf{g}_t\|}{\|\mathbf{g}_{\text{ref}}\|}$. $k_2 \geq 1$ holds for a larger proportion of all cases when the number of examples is smaller.

Figure 6: The average accuracy and execution time when the number of tasks is large.

When the number of training examples is small, we empirically show that it is more common that $k_2 > 1$. This explains why MEGA-I's update direction is dominated by the current gradient's direction while MEGA-II still allows adequate rotation. This property helps MEGA-II obtain better performance than MEGA-I when the number of examples is small.

We construct 20 tasks with $X$ number of examples per task, where $X = 200, 600$ and $55000$. The way to generate the tasks is the same as in **Permuted MNIST**, that is, a fixed random permutation of input pixels is applied to all the examples for a particular task. During the learning process, we record the norm of the gradient on the current task and the norm of the gradient on the episodic memory in each mini-batch.

In Figure 5, we use histogram to show the distribution of $\log(k_2)$ of MEGA-I. As we can see, when the number of examples per task is smaller, $k_2$ tends to be greater than 1 for a larger proportion. In particular, when the number of examples per task is 55000, 3.61% of all $k_2$ are less than 1 and when the number of examples per task is 600, 3.15% of all $k_2$ are less than 1. Notably, when the number of examples per task is 200, only 1.05% of all $k_2$ are less than 1. As explained in the last paragraph, if $k_2 > 1$, then $\theta_1 \leq \theta_2$, which means MEGA-II allows a more significant rotation of the current gradient. So MEGA-II can offer better performance than MEGA-I, especially when the number of examples is small.

### A.7.1 MEGA-2 Outperforms Other Baseline and MEGA-1 When the Number of Tasks is Large

In this section, we increase the number of tasks 30, 50 and 70. Each task have 200 examples and is constructed in a similar way to the **Permuted MNIST**. In Fig. 6, we show the average accuracy and execution time for all the methods and all the cases. We can see that the proposed MEGA-II outperforms all the baselines, except in the cases of 30 tasks. From the execution time comparison in Fig. 6(b), we can see that MEGA-II is much more efficient than MER [24]. Note that MEGA-II also significantly outperforms MEGA-I in this case.