[Reviews · NeurIPS 2020]

Review 1

Summary and Contributions: This paper introduces two episodic-memory based methods for lifelong learning that are shown empirically to outperform a host of existing state-of-the-art methods on standard lifelong learning tasks. The paper first provides a unified perspective on two prominent episodic-memory based methods, GEM and A-GEM, by characterising their updates as dynamically mixing the estimated gradients of the current task and those of the memory database. It uses this insight to motivate two new algorithms in this framework, that are more flexible in their weighting of the two gradients: (i) MEGA-I, which essentially balances the two gradients by the ratio of the loss on the current task to that of the memory database, and (ii) MEGA-II, which rotates the gradient on the current task to optimally align with that of the memory database before applying the loss balancing.

Strengths: The two MEGA methods are simple, intuitive and clearly described, and empirically show impressive results in a thorough comparison to many existing methods on benchmark lifelong learning tasks, using a number of metrics from the original A-GEM paper. The unified perspective of GEM and A-GEM as the balancing of two separate gradients is also a valuable contribution in and of itself, as well as providing an intuitive motivation for the MEGA methods.

Weaknesses: - The paper could benefit from some more insight into the workings of the two methods. For example, it would be interesting to see how the loss values on the current task and on the memory base change during training compared to other methods, to show how the loss-balancing affects the process of learning. - It is shown empirically that the ratio of the task gradient to the reference gradient is greater than one more often when the number of training examples is small, and this fact is used to explain why MEGA-I is more prone to catastrophic forgetting than MEGA-II in this case. Is it just the number of training examples that is limited in this scenario or is it also the number of gradient updates? Presumably it is the latter, because otherwise the training loss on the current task should converge faster and the ratio should spend less time above 1. This should be clarified. - Error bars are missing from many of the plots (Figures 2, 3, 4). Were experiments only run with one random seed per setting?

Correctness: Yes.

Clarity: Yes, the new algorithms are well motivated and explained, and the experimental results are clearly presented, though many plots do not have error bars (Figures 2, 3, 4). I think it would be better to include a table of hyperparameters in the appendix, rather than refer to the A-GEM paper.

Relation to Prior Work: Yes, the method is placed in context of the literature and, in particular, to two methods, GEM and A-GEM, to which it closely relates.

Reproducibility: Yes

Additional Feedback: *** POST- REBUTTAL COMMENTS *** Thank you to the authors for addressing some of my concerns, namely (i) adding error bars to the plots, (ii) clarifying how the number of training examples affects the number of gradient updates and (iii) providing plots showing how the loss on the memory and the current task change during training. Perhaps I was not entirely clear regarding (iii) - I think showing how the balance of the losses changes throughout training (i.e. the evolution of alpha_1 and alpha_2 over time) might give some nice insight into how the MEGA methods work. ***** - The random variables xi and zeta introduced in equation 1 are confusing. Don’t these just correspond to the data from the current task and the memory buffer? If so, why not write those explicitly? - Typos/grammar: Line 93, space before full stop. Suggestion for title: put a hyphen between memory and based, and perhaps omit the word algorithm. Line 167. “put” -> “puts” Line 254. “as” -> “to be” Line 254. “its” -> “their” Line 289. Remove “the two”


Review 2

Summary and Contributions: Update: My initial review was positive with three questions on: comparison wrt previous work, clarification on the upper bound/baseline, analysis of results. The author response addresses the question on the upper bound/baseline clearly, and my question on the analysis of results (Fig.4) turned out to be my misreading of the colours in the figure (MER vs A-GEM). The response however is not convincing on the question related to comparison. While the paper does have two important comparisons (A-GEM and GEM), it also needs to include more recent work on this topic, e.g., [36]. There are indeed differences wrt how the memory is chosen in [36], but at its core, this remains an episodic memory based approach. I am not suggesting a comparison with all the methods out there, but considering more recent and very relevant work is necessary. Hopefully, this can be addressed in the final version of the paper. In summary, I continue to support the paper for acceptance. The paper presents an episodic memory based approach for incremental or lifelong learning, where samples corresponding to tasks seen previously are stored and reused when adapting the model to new task(s). In particular, the proposed work is a generalization of two previous method: GEM and A-GEM. The core idea is to modulate the gradients based on the loss accumulated over the previously-seen as well as the new task. Two approaches for this, named as MEGA-I and MEGA-II, are presented in the paper. These methods are evaluated on standard benchmark datasets popular in this topic, and are also compared extensively wrt A-GEM/GEM, and also a few recent methods. Some ablation studies is also provided to help analyze the performance. Overall, the paper is well-written, and describes the approach clearly with sufficient detail. Source code included with the submission is another plus.

Strengths: * The paper is well-written, with a good description of the problem, the method, and its relation to previous work, especially A-GEM and GEM. * There is sufficient technical detail in the paper, which helps its reproducibility. Further, source code provided with the submission is another plus. * The experimental protocol is fairly clear, follows previous work quite closely, which makes the comparison easy to replicate.

Weaknesses: My main questions/concerns are on the empirical evaluation. * The choice of methods considered for comparison is unclear, except of course, A-GEM and GEM, which the method is very closely related to. For a fairer picture of the topic, other more recent methods should also be included, e.g., [30] or [36]. * The multi-task baseline (lines 244-245) appears a tad strange. How can this be a realistic upper bound performance? Firstly, this baseline needs to be explained clearly in the paper. Secondly, it is unclear why the proposed method performs better than this, but may be addressing the first point will clarify this. * The paper is missing an important baseline---the performance when all the tasks are learned jointly. I would consider this as a real upper bound performance. I have not seen this, unless I missed something. * Figure 4 also shows a curious behavior. The performance of A-GEM is either slightly better than or very similar to MEGA-I. Why is this the case? This calls for a closer examination of the \alpha's in cases with limited data.

Correctness: Yes, barring a few additions/clarifications mentioned above.

Clarity: Yes

Relation to Prior Work: Yes

Reproducibility: Yes

Additional Feedback: * The philosophies behind MEGA-I and MEGA-II are quite different. One works directly with the losses (as shown in (6)), while the second one computes the angle between the gradients of the losses. This distinction and the motivation behind it could be explained more clearly in the paper. On a related note, I think presenting MEGA-II also through \alpha_1 and \alpha_2, as was done for MEGA-I, would lead to an easier flow of thoughts. * MEGA-II would also be easier to understand visually, where one shows the gradient vectors and the angles schematically. * Minor comment: Not sure if Section 3 needs to standalone. It could be combined with Section 4.


Review 3

Summary and Contributions: This paper deals with the question of continual or lifelong learning in neural networks. There has been a plethora of recent and historical work on this topic, finding different ways to help networks alleviate the issue of catastrophic forgetting --- where a network trained on tasks A_0 through A_i, forgets these to differing degrees when trained on tasks A_i+1 onward. Most methods can be divided into regularisation based, memory based or meta-learning based. One relatively recent work is GEM (gradient of episodic memory) (and relatedly A-GEM). This works by storing examples from seen tasks in an episodic memory. When learning a new task, the gradient update is modified such that it does not increase the loss on examples from previous tasks (these are represented by the examples in memory). This work seeks to generalise the approach taken in works like GEM. The argument made is that GEM does not dynamically decide how much to care about the current task versus previous tasks. Mathematically, there is a contribution to the update on the weights in GEM from the gradient of the loss on the current task and from the memory. The argument made is that in GEM the "coefficient" before the contribution to the gradient update from the current task is set as 1. The paper proposes two alternate solution MEGA-I and MEGA-II that dynamically adapt these coefficients. MEGA-I is relatively straightforward conceptually. If the loss on the current task is less than some epsilon, we make the coefficient before the update from the current task 0 and we focus on stabilizing performance on the old tasks (ie the memory gradient gets priority). This method is shown to work surprisingly well. One complaint would be that this feels very much like a heuristic as opposed to a well-motivated design choice. MEGA-II is a bit more involved. I think the section describing this could be expanded a little to provide more clarity and detail on the derivations. The main motivation here is to find a rotated gradient on the current task such that it aligns well with both the current task loss and the old task loss. There is a derivation for how to then calculate the coefficients on each component of the gradient. In the extreme cases of old task loss = 0 or new task loss = 0 this prioritizes the other completely. The results presented look very impressive. One general compaint I have that is not specific to this paper, is that permuted MNIST and it's counterparts are not very good ways for us to measure performance in continual learning. It is an unfortunate choice that has become very popular but this dataset has now outlived its usefulness. There are also huge variations in performance based on how the dataset is used, how baselines are tuned, how many examples are used, hyperparameter tuning, the size of the model etc. It is unrealistic to assume reviewers or the field in general is able to verify that every method conforms to this strictly and inevitably there are differences. On a more cynical note then it is not surprising that so many methods consistently claim to improve the state of the art. We must move towards larger and more useful tasks. That aside, this is not a fault with the paper but the field. Given the datasets available the experiments are extensive and very conclusive. The results are very good. Especially surprising is the performance on CUB where this beats the multitask baseline. It would be great to see some explanation of how and why this happens or is possible.

Strengths: I've inlined all the comments above.

Weaknesses: I've inlined all the comments above.

Correctness: To the best of my knowledge, yes. But I have not run the code.

Clarity: The paper is well written.

Relation to Prior Work: In general, yes. There are a few memory papers missing from the discussion (eg Memory Networks, Weston et al and the literature around episodic memory including DNC, Neural Episodic Control). I think there should be a small discussion about episodic memory modules and also other papers that use memory for meta-learning/one-shot learning (Matching Nets was covered, but for eg "One-shot learning with memory-augmented neural networks" was not), specifically for continual learning (examples are MbPA -- Memory-based parameter adaptation, MbPA++, https://arxiv.org/pdf/1802.10542.pdf, https://papers.nips.cc/paper/9471-episodic-memory-in-lifelong-language-learning.pdf). Also there are some other examples of using cosines similarity or other methods related to gradients to weight different objectives as in "gradient surgery" https://arxiv.org/abs/2001.06782 or https://arxiv.org/pdf/1812.02224.pdf).

Reproducibility: Yes

Additional Feedback:


Review 4

Summary and Contributions: This paper first introduced a unified view of two episodic-memory-based algorithms, GEM and A-GEM, and then suggested two improved lifelong learning algorithms, MEGA-I and MEGA-II. According to the unified view of GEM and A-GEM, the two algorithms' final gradient is the weighted sum of the gradient of the current task and the gradient of episodic memory (previous tasks), with the weight alpha_1 and alpha_2. The authors show that both algorithms have alpha_1 = 1, arguing that it is a less robust option. To tackle this problem, the authors proposed two algorithms, MEGA-I and MEGA-II, where alpha_1 can have the value not only one but also non-one value. Two proposed algorithms outperform the comparative models, including GEM and A-GEM. Notably, this paper suggests one view to explain two previous algorithms and two newly proposed algorithms. The proposed algorithms also achieve state-of-the-art performance among 3 of 4 dataset w/o permuted MNIST. However, Some comparative models are missing in some experiments. GEM is not compared in Split CUB and Split AWA, though GEM is a main comparative algorithm. Other than that, PI is not compared in permuted MNIST and Split CIFAR. MER and PROG_NN are also not compared in Split AWA. The authors explain that they failed to run PROG_NN because of a memory issue, but a small round of the task (e.g., from 1 to 6 tasks) could be tested until memory capacity allows.

Strengths: Soundness of claims: The proposed view is plausible. Empirical evaluation: The proposed algorithms make competitive performance on lifelong learning benchmark. Significance and Novelty: Suggests two plausible lifelong learning algorithms making SOTA performance Relevance to the NeurIPS community: Many researchers studying continual learning and related field may be interested in this research.

Weaknesses: Theoretical Grounding: The proposed view may not cover all the lifelong learning algorithm using episodic memory. The argument for proposing MEGA-I and MEGA-II is relatively weak (e.g., arguing alpha_1 != 1, or newly proposed weight). Empirical Evaluation: some comparative algorithms are missing in the experiment of some datasets.

Correctness: I did not find any problem with correctness.

Clarity: I did not find any problem with clarity.

Relation to Prior Work: I did not find any problem with literature survey.

Reproducibility: Yes

Additional Feedback: line 112: it seems D^{te}_{t} → D^{te}_{t-1} Eq. (2), Eq. (8): may write \argmin_w using the instructions written below [https://tex.stackexchange.com/questions/5223/command-for-argmin-or-argmax](https://tex.stackexchange.com/questions/5223/command-for-argmin-or-argmax) Is there some explanation why "MEGA-I w/ alpha_1 = 1 and alpha_2 = 1" in table 1 do better than GEM in 2? The author argued that one of the important issues of GEM and A-GEM is alpha_1 = 1. ------------------------------------------------------------------------------------------ A response after the rebuttal period: Thank you for responding the review. I understood the paper more, thanks to other reviewers' reviews and authors' rebuttal. The new result on Split AWA somewhat alleviated my concern, and I raised my score. ------------------------------------------------------------------------------------------

[Author Response · NeurIPS 2020]

Thanks for all the valuable comments. Please check our responses below. We will address all minor comments.

**R1 Q1: how the loss on the current task and on the memory change during training compared to other methods.**

**A1:** In Figure 1b and 1c, the losses on the memory and on the task when the model is trained on the second task are
plotted. On the task, all the methods share similar noisy pattern. On the memory, MEGA-I and MEGA-II achieve
smaller error compared with A-GEM which reveals why MEGA-I and MEGA-II overcome forgetting.

**R1 Q2: Is it the number of training examples that is limited or is it also the number of gradient updates?**

**A2:** We agree. When the number of training examples is limited, if we fix the batch size, the consequence is that the
number of gradient updates is smaller. The training loss converges slower and the ratio spends more time above 1, as
the reviewer pointed out. This is also consistent with our derivation and empirical results in Appendix A.7. We will
make it more clear in the revised version.

**R1 Q3: Error bars.**

**A**: The results are averaged across 5 runs with different random seeds. In Figure 1a, for an example we add error bars
for the plot on MNIST and we will add all the missing error bars in the revised draft.

(a)        (b)        (c)

**R1 Q4: The random variables xi and zeta are confusing. Typos/grammar. Include table of hyperparameters.**

**A:**Thanks for the suggestions. We will fix them in the revised version.

**R2 Q1: For a fairer picture of the topic, other more recent methods should also be included, e.g., [30] or [36].**

**A**: Thanks for the comments. The paper [30] focuses on task-free continual learning which is a different setting from
ours. In [36], the authors focus on sample selection for episodic memory based lifelong methods which is an orthogonal
topic. In this paper, we use the same sample selection method (uniform sampling) for all episodic memory based
lifelong learning methods for a fair comparison which is also the strategy employed in A-GEM [2].

**R2 Q2: The multi-task baseline.**

**A**: In the multi-task baseline, all the tasks are learned jointly (i.e., the examples of all the tasks are shuffled and the
model is optimized over a single pass over the examples with SGD). The proposed methods only perform better than
the multi-task baseline on the *Split CUB*. This can be possibly attributed to the joint effects of episodic memory and
better optimization algorithms. First, by storing examples in the episodic memory, the examples of old tasks can be
accessed *multiple times* instead of just *one time* as in the multi-task baseline. On the other hand, with the proposed
better balancing schemes, the proposed methods outperform GEM and A-GEM, and also surpass the multi-task baseline
on the *Split CUB*.

**R2 Q3: Missing an important baseline—the performance when all the tasks are learned jointly.**

**A**: Thanks for pointing this. The missing baseline is the multi-task baseline used in the paper. We will clarify this.

**R2 Q4: Figure 4 also shows a curious behavior.**

**A**: In Figure 4, we can observe MEGA-I significantly outperform A-GEM (71% vs. 63% with 600 examples per task).

**R2 Q5: Additional feedback.**

**A**: We mentioned the derivation of MEGA-II through $\alpha_1$ and $\alpha_2$, at line 202-203. Please refer to Appendix A.3 for
details. And for the visualization of MEGA-II, we will add one figure to better illustrate the intuition.

**R3 Q1: Especially surprising is the performance on CUB where this beats the multitask baseline.**

**A**: Please refer to R2 Q2.

**R3 Q2: Here are a few memory papers missing from the discussion.**

**A**: We will add the missing citations and discussions.

**R4 Q1: Theoretical Grounding and Empirical Evaluation.**

**A**: The proposed view is used to point out one limitation of GEM and A-GEM which are the state-of-the-art lifelong
learning methods. The proposed methods are motivated by the fact that GEM and A-GEM always choose $\alpha_1 = 1$. The
results show the benefits of adjusting $\alpha_1$ during training. It is shown in [2] that A-GEM has better or comparable
performance than GEM, so we focus on comparing with A-GEM. We manage to finish the experiments of GEM,
PROG-NN and MER on *Split AWA* dataset (20 tasks), and the results are (GEM: 44.7±2.47, PROG-NN: 41.34±4.03,
MER: 36.34±3.94), all are inferior to MEGA-I (54.82±4.97). We will include all the results in the revised version.

**R4 Q2: Why "MEGA-I with $\alpha_1 = 1$ and $\alpha_2 = 1$" in table 1 do better than AGEM in 2. Typo/format.**

**A**: As shown in Figure. 6 in A-GEM [2], the angle between the task gradient and the reference gradient is mostly acute
(3500 / 5500), thus in A-GEM, $\alpha_2 = 0$ (as shown in Equation (4)) occurs frequently. In this case, the update totally
ignores the reference memory (leads to forgetting). In contrast, MEGA-I with $\alpha_1 = 1$ and $\alpha_2 = 1$ considers both the
reference memory and the current task in *each* update step which could better alleviate forgetting (although not as good
as MEGA-I which dynamically adjusts $\alpha_1$ and $\alpha_2$). We will address the format issue in the revised draft.

[Meta-Review · NeurIPS 2020]

The paper introduces a clear, simple generalisation of two established continual learning methods (GEM and A-GEM) which performs very well in a thorough empirical evaluation. All reviewers and the AC value the effort that the authors put in their response. There is consensus that the work has merit and all reviewers recommend accepting the paper (R1 and R4 raised their score).